

# Characterising evapotranspiration signatures for improved behavioural insights

Hansini Gardiya Weligamage[1], Keirnan Fowler[1], Margarita Saft[2], Tim Peterson[3], Dongryeol Ryu[1], and Murray C Peel[1]

[1]Department of Infrastructure Engineering, The University of Melbourne, Parkville, Victoria, 3052, Australia
[2]Institute of Applied Geosciences, Technische Universität Berlin, 10587 Berlin, Deutschland, Germany
[3]Department of Civil Engineering, Monash University, Clayton, Victoria, 3168, Australia

*Correspondence to*: Hansini Gardiya Weligamage (h.gardiyaweligamage@unimelb.edu.au)

**Abstract.** Hydrological signatures are statistical metrics useful to quantify and infer behaviours of hydrological processes, but there has been limited use of signatures for non-streamflow variables, such as actual evapotranspiration (AET). AET signatures can assist in tasks such as evaluating remotely sensed products, diagnosing deficiencies in hydrological models, and improving understanding of hydrological processes, such as the role of AET in driving hydrological drought. This study proposes eight AET signatures defined at various temporal scales from daily to annual. We demonstrate the value of AET signatures by using them to assess two remotely sensed AET (AET$_{RS}$) products against flux tower AET (AET$_{Fluxtower}$) at seventeen FluxNET sites in Australia. The two AET$_{RS}$ products are Moderate Resolution Imaging Spectroradiometer (MODIS, 16A2GFv06.1), and CSIRO MODIS Reflectance-based Scaling Evapotranspiration (CMRSET). Annually, median AET$_{RS}$ closely matches AET$_{Fluxtower}$, except in less-arid regions. However, signatures reveal RS$_{AET}$ largely underestimates the variability of flux tower data at both annual and monthly scales. Other monthly indices are better matched, such as indices of water stress and AET asynchronicity with potential evapotranspiration. However, some metrics are better matched in one product than the other, such as the strength and timing of seasonal fluctuations, with MODIS exhibiting a phase shift. Overall, the signatures reveal that regionally-developed CMRSET outperformed globally-developed MOD16A2GFv061. This study, the first to systematically define AET signatures, offers a way of assessing various aspects of AET dynamics across temporal scales. Furthermore, the case study highlights specific deficiencies in AET$_{RS}$ and may assist in selecting appropriate AET$_{RS}$, including for modelling studies.

## 1 Introduction

Hydrological signatures are statistical metrics used to quantify hydrological behaviours in catchments and can be used to compare hydrological behaviour across space (Addor et al., 2018; McMillan, 2021) and to assess the behavioural fidelity of hydrological model simulations against observations (Gupta et al., 2008). Over the past decade or so, there has been rapid advancement in the application of hydrological signatures, primarily calculated based on streamflow data and related to such categories as flow magnitude, duration, frequency, timing, and rate of change (Olden & Poff, 2003). In modelling, the use of signatures contrasts with commonly used aggregate metrics (e.g., Nash Sutcliffe Efficiency (NSE) or Kling Gupta Efficiency





(KGE)), which condense the information of coherence/discrepancy between two timeseries down to a single number. In contrast, signatures retain more detailed information on different (and ideally independent) aspects of the flow regime and may

be used to quantify model performance to each separate aspect. Likewise, the associated hydrological processes responsible for each aspect can be separately characterised. In practice, signatures have been widely used in modelling studies (Araki et al., 2022; Kiraz et al., 2023; McMillan, 2021; Westerberg et al., 2011) , while their linking to specific hydrological processes remains an open research question, with a key challenge being the interactions among different processes to produce emergent patterns in observed data (McMillan, 2020).

Although there has been a wide range of hydrological signatures defined for streamflow (McMillan, 2021; Olden & Poff, 2003; Safeeq & Hunsaker, 2016), signatures directly calculated based on other hydrological processes are rare, with some notable exception of studies that used the soil moisture and groundwater signatures (e.g., Araki et al., 2022; Heudorfer et al., 2019). Therefore, in this study, we are concerned with signatures of actual evapotranspiration (AET), which have not previously been researched to our knowledge. This lack of attention to AET signatures is surprising given the importance of AET in the overall

water cycle, comprising around 60% of the global terrestrial hydrological cycle (Abbott et al., 2019; Teluguntla et al., 2013). While there is literature investigating and quantifying AET in different study areas, examining at different spatial scales such as in-situ level (e.g., Rungee et al., 2019), grid level (i.e., remote sensing, e.g., Zhang et al.,  2010), catchment level (e.g., Avanzi et al., 2020), and regional level (e.g., Gardiya Weligamage et al., 2023), systematic and comprehensive studies focusing on AET signatures remain elusive. Moreover, some studies have used streamflow-based signatures such as total runoff ratio

(e.g., McMillan et al., 2014; Safeeq & Hunsaker, 2016), streamflow seasonality (e.g., Wrede et al., 2015), and diurnal cycles in streamflow (e.g., Schwab et al., 2016; Wondzell et al., 2010) to examine AET processes, although these signatures are only indirectly related to those AET processes. Furthermore, McMillan (2020) confirms that none of these streamflow-based signatures have investigated AET processes at shorter temporal scales, such as event scale.

We envisage at least three potential uses for AET signatures, namely 1) assessing the quality of remotely sensed AET products,

2) diagnosing deficiencies in hydrological models, and 3) improving understanding of hydrological processes. Assessment of the quality of remotely sensed AET products is important as these products are widely used across many research areas due to their ability to provide mostly continuous spatiotemporal data, unlike flux tower measurements (Yan et al., 2018; Zhang et al., 2016). However, their capacity to accurately predict various aspects of AET behaviour is often minimally assessed when incorporating them into a modelling study, and AET signatures can provide a more informative assessment. Secondly, AET

signatures can be employed to diagnose deficiencies in hydrological models, as the poor representation of AET could significantly impact streamflow predictions, particularly under changing climatic conditions such as climate change or multiyear droughts (Araki et al., 2022; Koster & Suarez, 2001; Peterson & Fulton, 2019). Thirdly, as noted by McMillan (2020), signatures permit extraction of "meaningful information about watershed processes", and it is often possible to define signatures to specifically provide information about a process of interest. This is particularly relevant since AET is the second

largest water balance component globally (after precipitation). Moreover, recent studies suggest that AET is a contributing





cause to changes in rainfall-runoff relationship for the same streamflow under multiyear drought at both annual and seasonal scales (Gardiya Weligamage et al., 2023; Peterson et al., 2021).

Highlighting the importance of using AET signatures, in this paper, we define a set of AET signatures and demonstrate their use in one of the three specific contexts listed above, namely the evaluation of remotely sensed products of AET. We define

distinct AET signatures for various temporal scales to best capture AET characteristics relevant to each timescale. Using AET signatures, two remotely sensed products are evaluated against flux tower data at several sites covering different climatic regions in Australia.

## 2 Material and Methods

Given that the primary purpose of this paper is to introduce a set of signatures for actual evapotranspiration, we begin this

section by describing and defining the signatures themselves. This is followed by descriptions relevant to the case study, including the study area, data, and specific methods to utilise the signatures in this case.

### 2.1 Proposed set of signatures

We propose eight signatures to quantify AET behaviour. We do not consider this an exhaustive list but seek a set that is reasonably representative of the variety of characteristics inherent to AET dynamics as a good starting point for future research.

As with streamflow signatures, the metrics cover a wide range of timescales, and different metrics require different temporal aggregations of AET information (specifically daily, monthly, or annual) tailored to the metric in question.

At the annual scale, we select two signatures, namely, the *long-term median* ($\overline{AET}_{annual}$) and interannual variability, expressed as the *coefficient of variation of annual AET* ($CV_{annual}$). We adopt $CV$ rather than absolute variability measures (e.g., variance, standard deviation, or interquartile range) deliberately following streamflow signature studies such as (Clausen

& Biggs, 2000), as it facilitates the comparison of variability across sites with different means.

To quantify seasonal variations, we adopt two signatures: a measure of *periodicity* ($P_{12month}$) and a measure of the *timing of the seasonal peak* ($TSP$) in AET. The $P_{12month}$ in this study quantifies the tendency for AET variation to recur with the seasonal cycle, i.e., with a period of 12 months, and as such, is calculated as the lag-12 autocorrelation of monthly AET. This will be unity in cases where the timeseries data varies with a perfectly repeating seasonal cycle. The $TSP$ is determined from

monthly timestep data by examining the median AET for each of the 12 calendar months and identifying the month with the maximum median AET. Note that the selection of median over mean AET is intended to minimize the influence of extreme values.

At the monthly scale, the monthly variability is quantified using the *monthly CV* ($CV_{monthly}$). Moreover, we define two signatures related to water stress, with the idea that the absence of water stress leads to AET perfectly mimicking potential

evapotranspiration (PET). When AET deviates from PET, it marks a deficit in water availability, which may be temporary (e.g., seasonal) or prolonged (as would be seen in arid environments). Both these water stress indices assume that the user has





access to a timeseries of estimated PET. The first such water stress metric, the *water stress* ($WS$) is defined as the difference between average monthly PET and average monthly AET, divided by the average monthly PET. Thus, higher values of this water stress signature indicate less rainfall and/or high PET. Initial testing revealed this metric to be very sensitive to aridity,

since it is obvious that AET will be much less than PET in arid areas. In non-arid areas, there might arise temporary (seasonal) water deficits that are not well characterised by the first water stress metric; thus, a subsequent metric was defined to these temporary fluctuations in water stress. This second water stress signature is purely based on the asynchronicity between normalised PET and AET, and thus is referred to as '*AET asynchronicity to PET*' ($AAP$). This latter signature can be thought of as quantifying the area between the normalised curves. It is calculated as follows. PET and AET monthly timeseries are first

normalised by dividing them by their mean monthly PET and AET values respectively. Then, the numerator is calculated by integrating the absolute difference between normalised monthly PET and AET ($\int(|normPET_{month} - normAET_{month}|)\,dt$). The denominator of the metric is then calculated by summing the maximum value among normalised PET and AET at each monthly time step ($\int \max(normPET_{month}, normAET_{month})\,dt$). The rationale behind calculating the absolute difference between values is that a difference between the curves is indicative of asynchronicity, regardless of which of the curves happens

to be greater at the given point in time.

In addition to annual, seasonal and monthly dynamics, the event scale is also important, even though it is assessable only for certain data types (e.g., flux tower data; simulations from daily timestep models) and not others (e.g., remotely sensed information provided on timesteps greater than a day). To assess AET dynamics at the event scale, we explored several options to quantify AET responsiveness to a rainfall event – in other words, the degree to which a rainfall event causes a jump in AET.

However, the difficulty of such a metric is that rainfall events may not only influence AET on the given day but also influence AET in the following days. If so, a standard correlation measure would be insufficient, but a lagged correlation is difficult to define since we do not know the lag a priori (and it may change over time). Seeking a generalisable metric, we select rainfall events greater than a threshold and identified the maximum daily AET value after the given rainfall event, up to a certain window duration (in days) after that event. We then apply a standard linear correlation equation to the anomalies of these

ordered pairs of numbers (i.e., rainfall anomaly versus maximum AET anomaly in the window after the rainfall event). Since this formulation is different from commonly used correlation metrics, we call it simply the '*Index of AET responsiveness to a rainfall event*' ($R$). For the purposes of the demonstration, we subjectively set the rainfall threshold and window duration parameters as 5 mm/day and 10 days, respectively. It is noted that this window size is also sensitive to the gap between selected rainfall events. If two rainfall events exceeding 5mm/day occurred within the 10-day window period, the window size is

restricted to the days between the two rainfall events, and the maximum AET value was chosen from that restricted window. Table 1 summarises the proposed eight signatures.

## 2.2 Study Area

Relative to global averages, Australia is a dry continent with annual average precipitation below 450 mm/year (Isaac et al., 2017). However, the coastal areas from southeast Australia to northern Australia receive comparatively higher precipitation,





exceeding 1000 mm/year in many areas. This study focuses on seventeen OzFlux sites in Australia (Figure 1). OzFlux, a part

of the international FluxNET program, is a micrometeorological monitoring network in Australia and New Zealand equipped

**Table 1: AET signatures**

| Temporal scale | Signature | Mathematical formulation |
|---|---|---|
| Annual | 1. Long-term Median $(\widetilde{AET}_{annual})$ | *Median of annual AET* |
| | 2. Interannual Variability $(CV_{annual})$ | $CV_{annual} = \dfrac{\sigma_{AET_{annual}}}{\overline{AET}_{annual}}$ |
| Seasonal (calculated using monthly timestep data) | 3. Periodicity $(P_{12month})$ | *Lag 12 auto correlation of monthly AET* |
| | 4. Timing of Seasonal Peak $(TSP)$ | *month with maximum median of monthly AET* |
| Monthly | 5. Monthly Variability $(CV_{monthly})$ | $CV_{monthly} = \dfrac{\sigma_{AET_{month}}}{\overline{AET}_{month}}$ |
| | 6. Water Stress $(WS)$ | $WS = \dfrac{\overline{PET}_{month} - \overline{AET}_{month}}{\overline{PET}_{month}}$ |
| | 7. [a]AET asynchronicity to PET $(AAP)$ | $AAP = \dfrac{\int (\lvert normPET_{month} - normAET_{month}\rvert)\, dt}{\int \max(normPET_{month}, normAET_{month})\, dt}$ |
| Event-scale | 8. Index of AET responsiveness to a rainfall event $(R)$ | $R = \dfrac{\sum P_{anom_{event}} * AET_{anom_{event}}}{\sqrt{\sum P_{anom}{}^2 \sum AET_{anom}{}^2}}$ where, $P_{anom_{event}} = Rainfall_i - \overline{Rainfall_i}$ $AET_{anom_{event}} = AET_{i+j} - \overline{AET_{i+j}}$ $i$ is the position of the rainfall event, and j is the day with maximum AET after the rainfall event, $0 \le j \le 10$ |

[a]Note that the trapezoidal integration was conducted.

with eddy covariance measurement, providing information on carbon, energy, and water exchange. The seventeen sites

constitute the majority of flux towers in Australia; while seven other active flux towers exist, they were excluded due to

insufficient coverage (i.e., < 7 years) and considerable percentages of negative and unavailable data. The selected study sites

cover a wide range of climate and ecosystem regions in Australia, as summarised in Table 2. The time period of data availability

varies at each site (typically 7-20 years). Hence, different periods of data coverage are adopted in this study to perform the

analysis at each site.


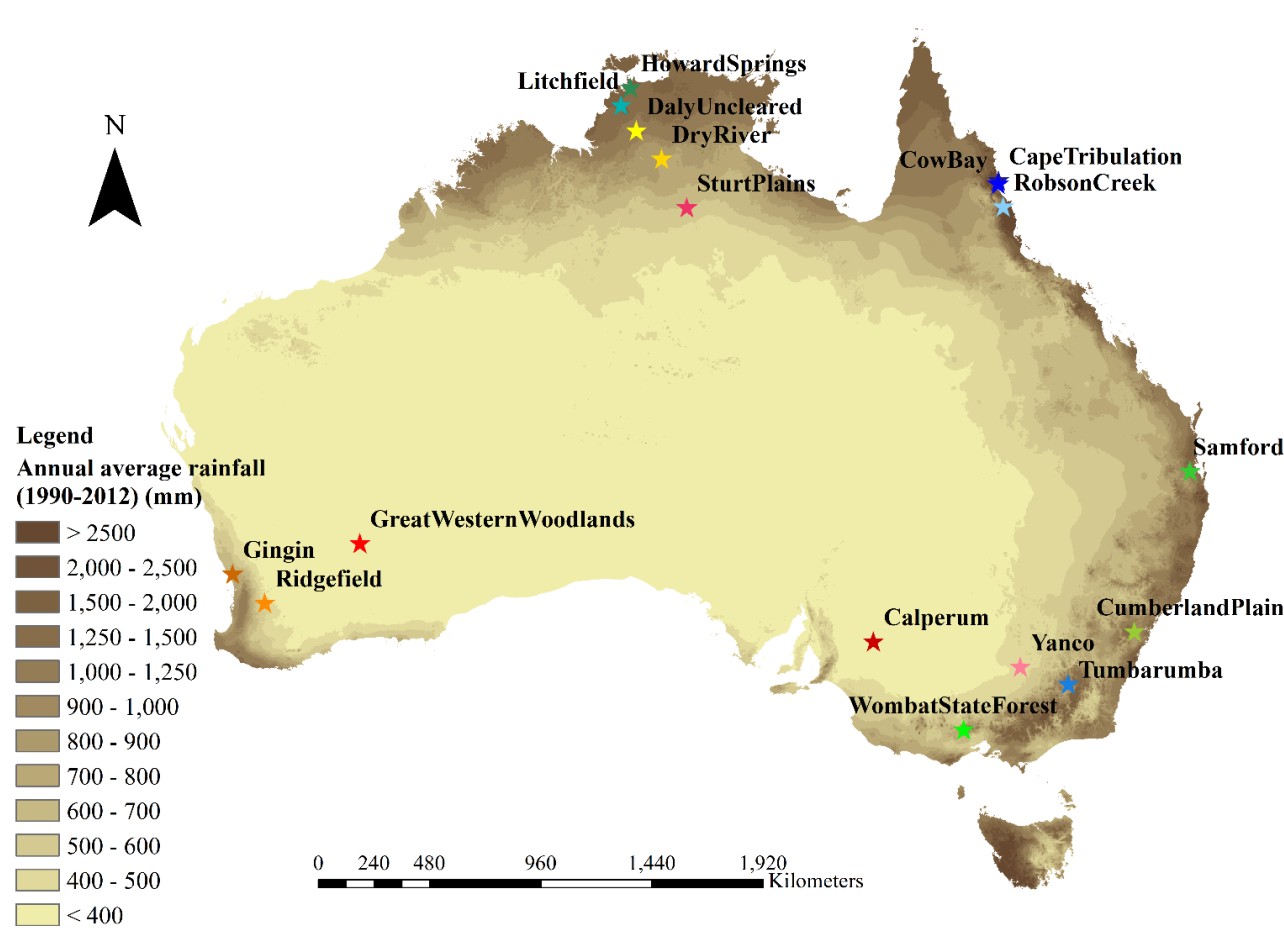

**Figure 1: Locations of selected OzFlux sites in this study.**

**Table 2: Summary of OzFlux sites**

| Site | Lon. | Lat. | Data coverage[a] | Köppen Climate[b] | Eco-region[b] | Arid-ity[c] | Data source |
|------|------|------|------------------|-------------------|---------------|-------------|-------------|
| Calperum | 140.587 | -34.003 | Jul-10 to Feb-24 | Arid desert cold | Mediterranean woodlands | 4.99 | (Meyer et al., 2024) |
| Great Western Woodland | 120.654 | -30.191 | Jan-13 to Jan-24 | Arid desert hot | Mediterranean woodlands | 4.80 | (Macfarlane et al., 2024) |
| Sturt Plains | 133.350 | -17.150 | Aug-08 to Feb-24 | Arid steppe hot | Tropical grasslands | 3.18 | (Beringer, Hutley, et al., 2024e) |





| Yanco | 146.290 | -34.987 | Jan-13 to Feb-24 | Arid steppe cold | Temperate grasslands | 3.14 | (Beringer, Walker, et al., 2024) |
|---|---|---|---|---|---|---|---|
| Ridgefield | 116.966 | -32.506 | Jan-16 to Feb-24 | Temperate dry, hot summer | Mediterranean forests woodlands and shrub | 2.95 | (Beringer, Lardner, et al., 2024) |
| Gingin | 115.713 | -31.376 | Oct-11 to Feb-24 | Temperate, dry, hot summer | Mediterranean woodlands | 2.29 | (Silberstein et al., 2024) |
| Dry River | 132.370 | -15.258 | Oct-09 to Feb-24 | Tropical savanna | Tropical savannas | 2.20 | (Beringer, Hutley, et al., 2024b) |
| Daly Uncleared | 131.388 | -14.159 | Jan-08 to Dec-23 | Tropical savanna | Tropical savannas | 1.75 | (Beringer, Hutley, et al., 2024a) |
| Cumberland Plains | 150.723 | -33.615 | Jan-14 to Dec-23 | Temperate, no dry season, hot summer | Temperate woodlands | 1.57 | (Pendall et al., 2024) |
| Samford | 152.877 | -27.388 | Jun-10 to Dec-17 | Temperate no dry season, hot summer | Temperate broadleaf and mixed forest | 1.36 | (Grace et al., 2024) |
| Wombat State Forest | 144.094 | -37.422 | Jan-10 to May-21 | Temperate, no dry season, warm summer | Temperate broadleaf forest | 1.20 | (Arndt et al., 2024) |
| Howard Springs | 131.152 | -12.495 | Jan-02 to Aug-22 | Tropical savanna | Tropical savannas | 1.17 | (Beringer, Hutley, et al., 2024c) |
| Litchfield | 130.794 | -13.179 | Jun-15 to Dec-23 | Tropical savanna | Tropical savannas | 1.17 | (Beringer, Hutley, et al., 2024d) |
| Robson Creek | 145.630 | -17.117 | Aug-13 to Dec-23 | Temperate, dry winter, hot summer | Tropical and sub-tropical moist broadleaf forests | 0.98 | (Liddell & Weigand, 2024c) |
| Tumbarumba | 148.151 | -35.656 | Jan-02 to Dec-22 | Temperate no dry season, warm summer | Temperate broadleaf and mixed forest | 0.89 | (Stol & Kitchen, 2024) |
| Cow Bay | 145.427 | -16.238 | Jan-09 to Jul-23 | Tropical rainforest | Tropical and sub-tropical moist broadleaf forests | 0.55 | (Liddell & Weigand, 2024b) |
| Cape Tribulation | 145.446 | -16.103 | Jan-10 to Nov-18 | Tropical rainforest | Tropical and sub-tropical moist broadleaf forests | 0.42 | (Liddell & Weigand, 2024a) |

[a]This study used version 1 of the year 2024 (2024_V1) of flux tower data from TERN data portal.





---

[b] Guerschman et al. (2022).

[c]Aridity refers to the aridity index defined as the ratio of potential evapotranspiration (PET) to precipitation (P).

## 2.3 Data

### 2.3.1 OzFlux eddy covariance evapotranspiration data

AET data, version 1 of the year 2024 (2024_V1), were obtained from OzFlux towers through the Terrestrial Ecosystem Research Network (TERN) data portal (https://portal.tern.org.au/, last accessed on 30/04/2024) at daily, monthly, and annual time scales. To minimize variability among sites due to different processing steps, the data providers consistently apply PyFluxPro (v3.4.17) to implement a standardized method to process data, as described in (Isaac et al., 2017). Level 6 flux tower data, adopted here, undergoes quality control, gap filling, and partitioning of the net ecosystem exchange of carbon

(NEE) data into gross primary production (GPP) and ecosystem respiration (ER). We conducted further data quality checks of the daily AET data and filtered out negative daily AET values. The maximum percentage of negative daily AET values among seventeen study sites was less than 2.3%.

### 2.3.2 Remotely sensed evapotranspiration data

Remotely sensed AET ($AET_{RS}$) products are popular due to the relative rarity of flux towers and because they provide spatially

distributed data, in contrast to flux tower data, which are near-point scale. Here, we present an example of the application of AET signatures to examine two $AET_{RS}$ products to assess their ability to capture different aspects of AET behaviours at various temporal scales. The two $AET_{RS}$ products are 1) Pixel resolution 500 m, gap filled, 8 days composites of version 6.1 of Moderate Resolution Imaging Spectroradiometer (MOD16A2GF.061) from Running et al. (2021), referred to as 'MODIS AET' hereafter, and 2) 30 m high-resolution, monthly composites of CSIRO MODIS Reflectance-based Scaling

Evapotranspiration (CMRSET) from McVicar et al. (2022). The MODIS AET product is one of the most widely used global AET datasets in hydrological and biogeochemical studies (e.g., Baker et al., 2021; Gaona et al., 2022; Salazar-Martínez et al., 2022), while CMRSET is an Australian regional product tested and used extensively over the corresponding region (e.g., Doody et al., 2023; Guerschman et al., 2022; Xu et al., 2022). In both cases, we extracted AET timeseries from pixels that contain the flux tower sites. The 8-day composites of MODIS AET data were aggregated into monthly data through weighted

temporal averaging.

### 2.3.3 Potential evapotranspiration data

As *water stress* and *AET asynchronicity to PET* require a timeseries of PET, they were quantified using monthly Morton's wet environment potential evapotranspiration (Morton, 1983) from the SILO (Scientific Information for Land Owners) database (https://www.longpaddock.qld.gov.au/silo/).





### 2.4 Method

To characterise the performance of $AET_{RS}$ products, the signatures described in Section 2.1 are calculated separately for flux towers AET ($AET_{Fluxtower}$) and $AET_{RS}$. Then, we investigated the deviation of $AET_{RS}$ signatures from $AET_{Fluxtower}$ signatures. As mentioned above, different periods have been considered at each flux tower site depending on their data coverage. For each site, the flux tower period of record determined the period of comparison between flux tower and remotely sensed information. Finally, the traditional efficiency metrics such as NSE, KGE, and components of KGE were calculated using monthly MODIS and CMRSET AET at each flux tower site in order to assess predictive skills and the challenges associated with using these traditional metrics compared to AET signatures.

## 3 Results

The AET signature results are presented in the four main categories based on the temporal scales in the order of 1) annual, 2) seasonal, 3) monthly, and 4) event-scale. To allow better contextualisation of AET signature results, we first present some example monthly timeseries plots of $AET_{Fluxtower}$, $AET_{RS}$, SILO rainfall, and PET at six flux towers (Figure 2). These timeseries plots highlight some aspects of AET that can be observed through visual inspections, such as AET variability, periodicity, and asynchronicity between AET and PET, which can subsequently be considered in AET signature results.

### 3.1 Annual AET signatures

Figure 3a shows an increase in the *long-term median* $(\widetilde{AET}_{annual})$ values with decreasing aridity index, as expected. For example, less arid sites (shown in blues), such as Cape Tribulation and Cow Bay, have higher $\widetilde{AET}_{annual}$, while more arid sites (shown in reds), such as Calperum and Great Western Woodland have lower $\widetilde{AET}_{annual}$. The comparison of signatures between flux tower and RS products shows that $AET_{RS}$ products tend to underestimate $\widetilde{AET}_{annual}$, except for less arid flux sites (i.e., aridity index <1 as in Table 2), such as Robson Creek, Cow Bay, and Cape Tribulation. Readers should bear in mind the difference in the spatial footprint of data (i.e., support), which is typically on the order of $10^2$-$10^4$ meters in the case of remotely sensed data (depending on the source, except high-resolution AET products developed based on AET sources such as Landsat and Sentinel-2), and approximately one order of magnitude smaller for flux towers (Chu et al., 2021; Finnigan, 2008). While some AET signatures may be relatively insensitive to this difference in the spatial footprint of data, it is reasonable to expect that the long-term average or median values may be sensitive to the position of the flux tower in the landscape, and thus, some variation in Figure 3a may be directly attributable to this.

Figure 3b shows a higher *coefficient of variation of annual AET* $(CV_{annual})$ at arid flux tower sites such as Calperum, Sturt Plains, and Yanco, whereas other sites show lower $CV_{annual}$ within the range of 0-0.2. $CV_{annual}$ is lower in both MODIS and CMRSET AET compared to flux towers at almost all flux towers, and the scatter is high, implying the representation of year-to-year variability is poor in MODIS and CMRSET compared to the actual inter-annual variability as seen in flux tower data.





**Figure 2: Monthly timeseries plots of flux tower and remotely sensed AET, rainfall and PET at six example flux tower sites.**

### 3.2 Seasonal AET signatures

Figure 4a shows *periodicity* ($P_{12month}$) via the lag-12 autocorrelation. No clear relationship is observed in $P_{12month}$ with aridity. For example, weaker $P_{12month}$ values are observed at both the more arid flux tower sites (e.g., Great Western Woodlands, Yanco) and less arid flux tower sites (e.g., Cape Tribulation). This result may reflect hydroclimatic factors since strongly seasonal climates can occur in both arid and temperate environments. Remotely sensed data performs more poorly in this metric than in any other. The $P_{12month}$ of CMRSET monthly AET shows significant scatter but minimal bias. However, MODIS monthly AET shows stronger periodic behaviour than ground measurements at most of the flux tower sites.

Figure 4b compares the *timing of seasonal peaks* (*TSP*) between $AET_{RS}$ and $AET_{Fluxtower}$ estimates. Here, CMRSET tends to show the same timing (7 out of 17 flux towers) or closer timing (e.g., one month offset - 6 out of 17 flux towers) of *TSP* as





flux towers at many flux tower sites, whereas those calculated using MODIS AET are significantly offset with flux tower *TSP*. This confirms that CMRSET tends to capture flux tower AET seasonal peaks more closely than MODIS AET. The MODIS results for these two metrics suggest that strongly periodic remote sensing estimates do not necessarily align well with the *TSP* in ground measurements (i.e., AET$_{Fluxtower}$).

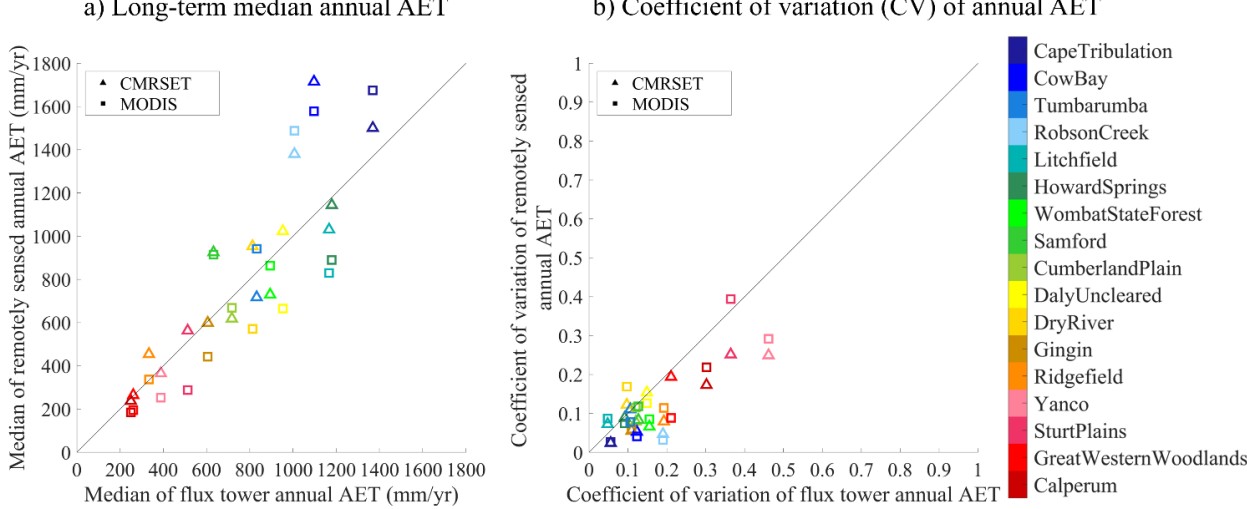

**Figure 3: Comparison of annual AET signatures; a) Long-term median AET, b) Coefficient of variation (CV) of annual AET, between remotely sensed AET (MODIS and CMRSET AET) and flux tower AET. (Note that the flux towers are ordered by aridity index – from Cape Tribulation (Lowest aridity) to Calperum (Highest aridity)).**

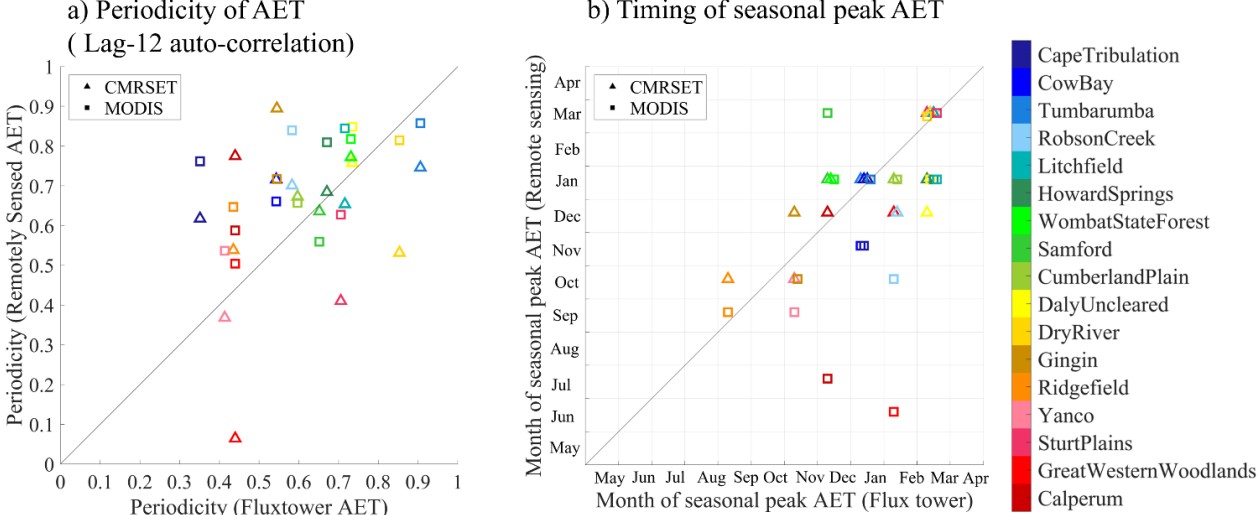

**Figure 4: Comparison of seasonal AET signatures, a) Periodicity of AET (Lag-12 auto-correlation), b) Timing of seasonal peak, between remotely sensed AET (MODIS and CMRSET AET) and flux tower AET. Note that Figure 4b values take integer values only (i.e., either one calendar month or the next), leading to several points overlying the same plotting position; to make every point visible we subject each point to a jitter (i.e., a unique offset within the same grid cell).**



### 3.3 Monthly AET signatures

Figure 5 quantifies and compares monthly AET signatures using $AET_{Fluxtower}$ and $AET_{RS}$. Figure 5a shows a lower $CV_{monthly}$ at less arid flux towers such as Cape Tribulation, Cow Bay, and Robson Creek, with most other flux towers grouping tightly between $CV$ values of 0.3 and 0.6, while Sturt Plains is a clear outlier at 0.9. $CV_{monthly}$ from CMRSET and MODIS AET do not show much overall bias relative to $AET_{Fluxtower}$, yet the match with observed is very poor, with considerable scatter due to overestimation and underestimation of $CV_{monthly}$, depending on site.

Regarding *water stress* ($WS$), Figure 5b shows how $WS$ increases with aridity, as expected. *WS* from CMRSET and MODIS AET is moderately well predicted except for wet flux towers, which are underestimated. It is possible that this is due to errors or biases in PET (recall that Morton's Wet Environment PET was used), which may be less of an issue when PET is clearly greater than AET, whereas in cases where they are similar PET errors may be more important. Figure 5c shows the *AET asynchronicity to PET* ($AAP$) that incorporates differences in phases between AET and PET timeseries. Results show that

both $AET_{Fluxtower}$ and $AET_{RS}$ are asynchronous with PET (i.e., $APP > 0$) for all the sites. However, at wetter sites, AET is more synchronous with PET showing smaller $APP$ values (i.e., smaller APP indicates closer synchronicity to PET), perhaps suggesting few temporal gaps in water availability, but increasing with aridity index (except at Calperum and Great Western Woodlands sites). Furthermore, the results show separation between remotely sensed products, with MODIS showing more *AAP* while CMRSET showing lower *AAP* compared to flux tower *AAP* for sites exhibiting higher than average asynchronicity.

### 3.4 Event-scale AET signatures

Figure 6 shows the index of AET responsiveness to a rainfall event ($R$). Recall that such information is unavailable for RS data due to its longer timestep. The $R$ of zero indicates no correlation between rainfall event and the subsequent AET, and Figure 6 shows no discernible correlation between the magnitude of a rainfall event (i.e., > 5mm/day in this example) and the subsequent AET on either the rain day or thereafter (i.e., maximum of 10 days window in this example) at the majority of the

flux tower sites, and even a slightly negative correlation, perhaps suggesting that rain days might be followed by cloudy weather that suppresses AET.

### 3.5 Traditional efficiency metrics

Figure 7 shows a range of commonly used efficiency metrics such as NSE, KGE, and the sub-components of KGE, namely, the ratio of standard deviations (α), the ratio of means (β), and the Pearson correlation coefficient (r), calculated using monthly

MODIS and CMRSET AET with flux tower AET. The distribution of NSE (median: -0.11) from MODIS AET indicates a poor prediction of $AET_{Fluxtower}$ on the monthly scale. In contrast, the distribution of NSE from CMRSET AET shows a positive, but still close to zero, median value of 0.14, implying better performance than MODIS but still overall poor performance.

For KGE, the median KGE values across flux tower sites for monthly MODIS and CMRSET AET are 0.45 and 0.53, respectively. The sub-components of KGE, such as α shows closer variability in MODIS AET (median α = 1.07) and in





CMRSET (median α = 0.94) compared to flux tower estimates. The β component of KGE shows a lower mean in MODIS AET (median β = 0.77) and a slightly higher mean in CMRSET (median β = 1. 07) compared to flux tower estimates. The r component of KGE shows a relatively similar correlation for both MODIS AET (median r = 0.77) and CMRSET (median r = 0.69) with $AET_{Fluxtower}$. Although these components of KGE provide valuable information about the ability of $AET_{RS}$ to capture dynamics from $AET_{Fluxtower}$, this information is diluted in the final KGE value. Figure S1 in the supplementary information shows the traditional efficiency metric values at each flux tower site.

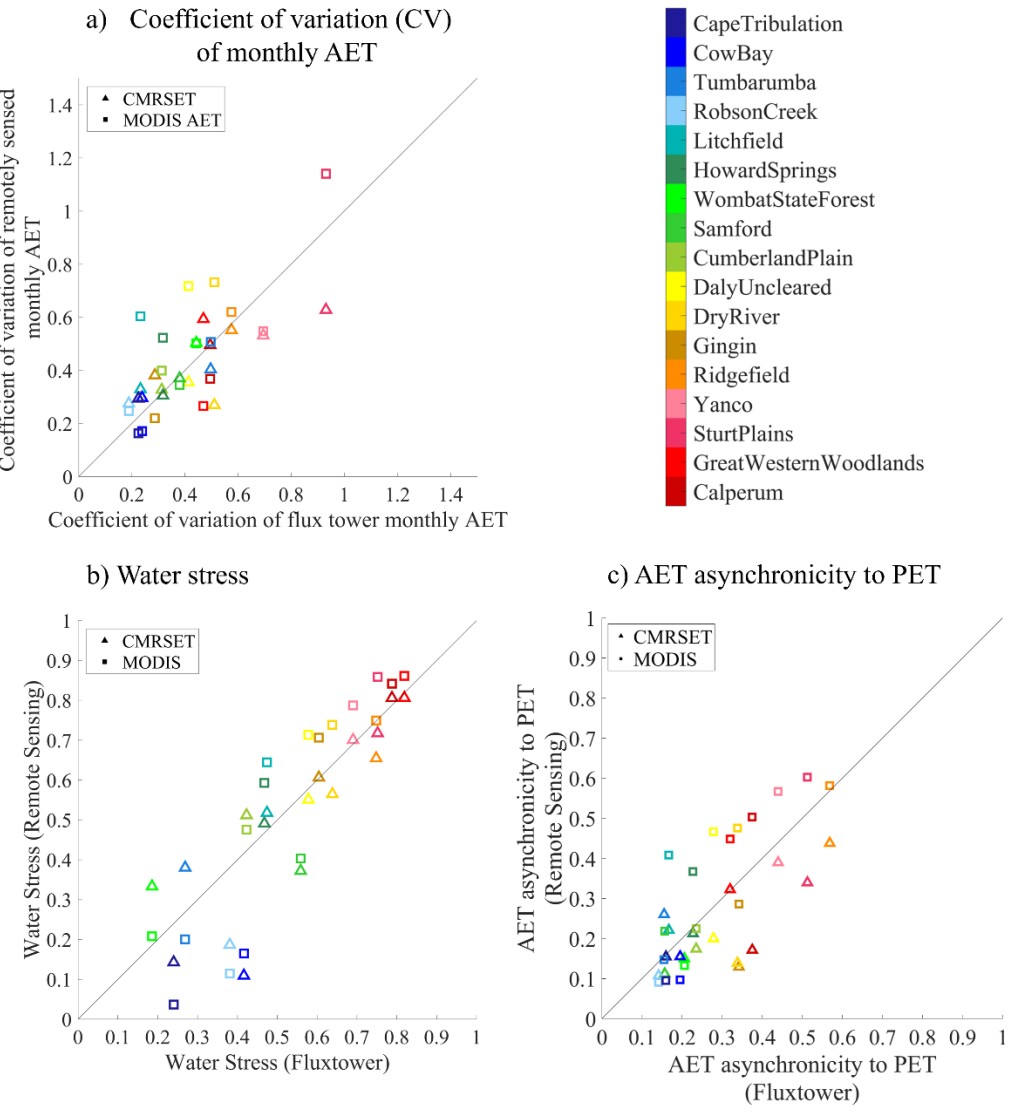

**Figure 5: Comparison of monthly signatures; a) Coefficient of variation of monthly AET, b) Water stress, and c) AET asynchronicity to PET, between remotely sensed AET (MODIS and CMRSET) and flux tower AET.**





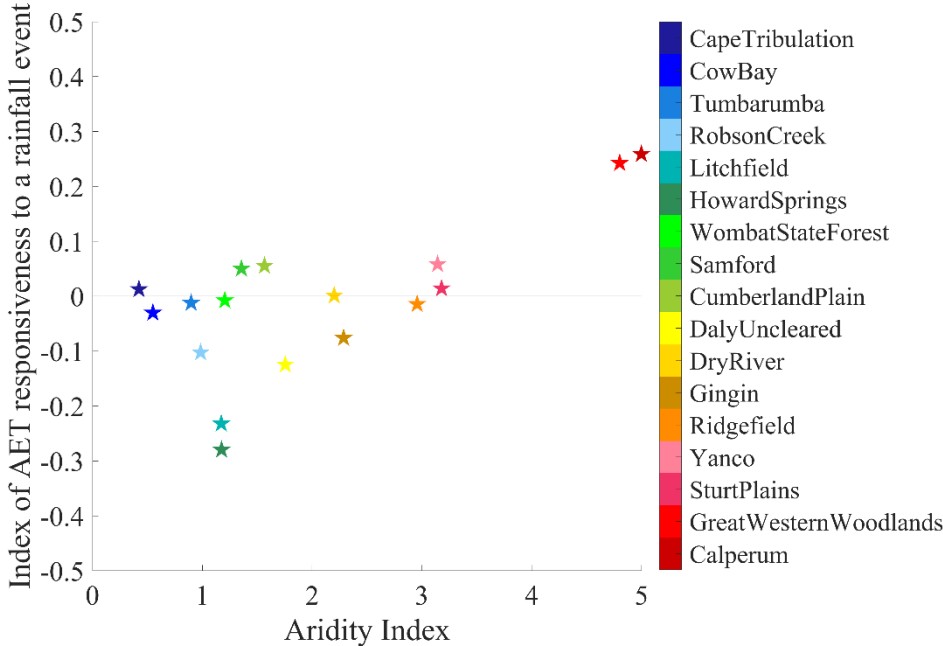


**Figure 6: Index of AET responsiveness to rainfall vs. aridity index at flux tower sites. (Note that the index of zero indicates no correlation between rainfall event and the subsequent AET).**

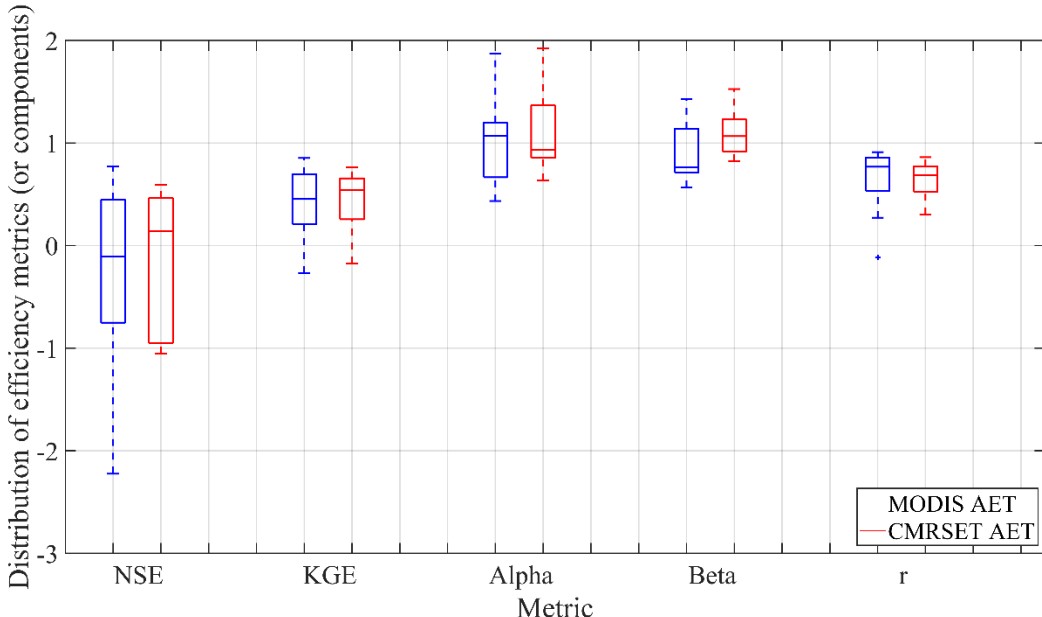

**Figure 7: Distribution of conventional efficiency metrics, all calculated on a monthly timestep: 1) Nash Sutcliffe efficiency (NSE), 2)**
**Kling Gupta Efficiency (KGE), 3) Alpha (α – ratio of standard deviations), 4) Beta (β – ratio of means), and r (r – Pearson correlation coefficient), calculated using monthly MODIS and CMRSET AET with flux tower AET (Note that there are two outliers not shown in the NSE calculated for MODIS and CMRSET, which are lower than -3).**



## 4 Discussion

This study developed evapotranspiration signatures at various temporal scales and used them to evaluate remotely sensed AET
information against flux towers. The study provides a basis for exploring what sort of signatures might be useful when
investigating and characterising AET behaviours, with applications across other domains, such as characterising catchment
processes and critiquing hydrological models.

### 4.1 Value of AET signatures over aggregate measures of performance

AET signatures introduced in this analysis offer comprehensive behavioural insights into AET to complement existing suites
of indices for streamflow and, to a lesser extent, groundwater. A key benefit of these signatures is their capacity to characterise
different aspects of AET dynamics, allowing the quantification of nuanced aspects that can be discerned through visual
inspection but, without signatures, are not easily defined numerically. Some examples are given in Table 3.

**Table 3: Example of aspects of AET dynamics that can be discerned via visual inspection of timeseries (Figure 2) and are
subsequently reflected in signature results (Figure 4 & 5)**

| Flux tower | Aspects of AET dynamics based on visual inspection | Corresponding signature results |
|---|---|---|
| Calperum | CMRSET closely synchronizes with PET variability | AET asynchronicity to PET($AAP$) is low (0.17; note a value of zero would mean perfect synchronicity) |
| | The seasonal variation in MODIS is shifted (peaks are too early) | Timing of seasonal peaks ($TSP$) indicates earlier peak (July for MODIS compared to December for CMRSET and fluxtower) |
| Gingin | The seasonal variation in fluxtower AET is offset with PET | Fluxtower $AAP$ is relatively higher (0.34) showing the asynchronicity with PET |
| | CMRSET closely synchronizes with PET variability. | CMRSET $AAP$ is low (0.12) |
| Cumberland Plain | Clear and regular seasonal cycle is observed in fluxtower and RS AET | Periodicity ($P_{12month}$) is greater than 0.6 for fluxtower and RS AET |
| | Fluxtower and RS AET are mostly synchronized | $TSP$ is same for MODIS and CMRSET (January) and only slightly different for fluxtower (February) |
| Wombat State Forest | Clear and regular seasonal cycle is observed in fluxtower and RS AET | $P_{12month}$ is greater than 0.73 for fluxtower and RS AET |
| | Clear and somewhat high temporal variability of monthly AET is observed in fluxtower and RS AET | Coefficient of variation ($CV$) is greater than 0.44 for fluxtower and RS AET |





| Howard Springs | Relatively high temporal variability of monthly AET is observed in fluxtower and RS AET | $CV$ is higher in MODIS (0.52) compared to fluxtower AET (0.32) and CMRSET (0.31) |
| | fluxtower and RS AET are mostly synchronized. | $TSP$ is same for MODIS and CMRSET (January) and different for fluxtower (March) |
| Robson Creek | Lower temporal variability of monthly AET is observed in fluxtower AET compared to RS AET | $CV$ is low in fluxtower AET (0.18) compared to CMRSET (0.27) and MODIS (0.24) |
| | RS AET is more responsive to and synchronizes with PET than fluxtower AET | Flux tower water stress ($WS$) is higher (0.38) compared to CMRSET (0.17) and MODIS (0.11) |


This capacity for nuanced characterisation in AET signatures contrasts with traditional fit/performance metrics such as NSE and KGE. As mentioned in the introduction, these commonly used performance metrics are often applied to quantify hydrological model performances but may obscure specific behavioural information (McMillan, 2021; Wagener & Gupta, 2005). For example, the NSE calculated using monthly MODIS and CMRSET AET at some flux tower sites showed negative

values, indicating bad predictive skills, whereas the KGE calculated using both the AET$_{RS}$ tended to show some predictive skills over flux tower observations. However, both of these conventional performance metrics fail to identify what aspects of AET in remote sensing products have led to poor prediction of AET$_{Fluxtower}$. Even though the subcomponents of KGE, such as the ratio of standard deviation, the ratio of mean, and the Pearson correlation coefficient, provided valuable information about the ability of these two AET$_{RS}$ products to capture AET dynamics compared to AET$_{Fluxtower}$, the final KGE value dilute this

information. Therefore, incorporating AET signatures into applications such as hydrological model calibration and evaluation could lead to more accurate and realistic outcomes.

**4.2 Insights into actual evapotranspiration behaviour at Australian flux tower sites**

This study confirms various anticipated AET behaviours at flux tower sites in Australia across different temporal scales. For example, the $\widehat{AET}_{annual}$ increased as the aridity index decreased as expected, indicating that less arid flux towers have a higher

$\widehat{AET}_{annual}$, whereas more arid flux towers have a lower $\widehat{AET}_{annual}$. However, the $CV_{annual}$ across flux tower sites was low (range of 0-0.2 at 14 out of 17 sites), regardless of their aridity index. This finding is consistent with the relatively constant annual AET variability over time, as reported by Gardiya Weligamage et al. (2023). In terms of $CV_{monthly}$ of flux tower AET, a higher degree of AET fluctuation was observed compared to the annual scale; however, flux towers situated in temperate climates with dry winters and hot summers (e.g., Cape Tribulation, Cow Bay, and Robson Creek) exhibited lower monthly

variability. As expected, $WS$ at flux towers largely increased with increased aridity. Regarding seasonal AET behaviour, the $TSP$ of flux tower AET was identified between November and March. Regarding the $P_{12month}$ of flux tower AET, no distinct relationship was found between periodicity and the aridity index. At the event scale, most flux towers did not show a discernible correlation between rainfall events and subsequent AET events. Therefore, at this stage, the '*index of AET responsiveness to*





*a rainfall event*' signature does not appear to add significant value in this AET signature analysis. However, we employed this

event-scale signature to evaluate commonly used conceptual rainfall-runoff model performances in a companion paper (Gardiya Weligamage et al., 2024), and the models were found to be very biased in this signature. Therefore, this event-scale signature adds value in constraining models or exploring model deficiency. Furthermore, this event-scale signature is particularly noteworthy, as it highlights a distinct aspect of AET dynamics not previously quantified. As McMillan, (2020) confirmed, no signature (even an indirect AET behaviour explanation using hydrological signatures) was previously identified

for assessing AET at the event scale.

### 4.3 Implications for quality of remotely sensed actual evapotranspiration

The eight AET signatures proposed in this study are used to highlight the strengths and weaknesses of two $AET_{RS}$ products (i.e., MODIS AET and CMRSET) on multiple aspects by assessing their AET signatures with those obtained from flux tower observations. Given that it seems increasingly common for studies (particularly modelling studies) to uncritically adopt $AET_{RS}$

products, these findings underscore the deficiencies in $AET_{RS}$ data and pinpoint several specific aspects of failure. Thus, we emphasize the importance of investigating $AET_{RS}$ data prior to use in any other applications, such as rainfall-runoff modelling. More specifically, most signatures, such as monthly and seasonal (except periodicity from MODIS monthly AET) AET signatures, reveal that $AET_{RS}$ is generally unbiased compared to $AET_{Fluxtower}$. However, the wide scatter observed in the AET signatures underscores the challenge of reproducing the dynamics captured by flux tower data. This wide scatter was seen in

several signatures such as $CV_{monthly}$, $AAP$, $P_{12month}$, and $TSP$. At the annual scale, RS-derived AET signatures of $\widehat{AET}_{annual}$ and $CV_{monthly}$ showed underestimation (negative bias) relative to flux towers. At the seasonal scale, challenges were particularly pronounced with the MODIS AET, which exhibited offsets in $TSP$, while being overly periodic compared to flux tower data. Therefore, caution should be exercised when extracting seasonal information from MODIS AET. Conversely, the regionally developed CMRSET product tended to align comparatively better with the $TSP$ observed by flux towers, likely because the

CMRSET model was calibrated only to flux tower sites in Australia (Guerschman et al., 2022), in contrast to MODIS which is globally calibrated and thus has less Australian focus. Our findings concur with Guerschman et al. (2022), who reported that the calibrated CMRSET model performs better than MODIS AET (i.e., MOD16A2). However, CMRSET exhibited greater scatter in terms of periodicity, highlighting issues with seasonal consistency.

### 4.4 Limitations and future studies

While this study assessed these signatures calculated using $AET_{Fluxtower}$ estimates and compared them with those calculated using $AET_{RS}$ estimates, there are uncertainties associated with flux measurement errors and spatial representativeness of both estimates. In this study, we assumed the $AET_{Fluxtower}$ to be robust and compared the AET signatures calculated using them with those calculated using remotely sensed products. However, these flux towers may be subjected to errors in energy closure in the eddy covariance method due to conditions such as weak turbulence or systematic flux leakage on hillslopes (Chen et al.,

2011; Wilson et al., 2002). Moreover, as mentioned, there can be differences in AET signatures due to the different spatial

footprints of observation between $AET_{RS}$ and $AET_{Fluxtower}$. Viewing this study as starting point for the study of AET signatures, future studies can expand or modify these signatures in order to best capture AET behaviours, including on other continents, and can be employed in hydrological model calibration and evaluation to represent AET processes realistically.

## 5. Conclusion

This study proposed eight AET signatures to explore AET behaviours at different temporal scales, expanding upon hydrological signature studies and providing comprehensive insights into AET dynamics. The AET signatures calculated from flux towers in Australia were consistent with anticipated AET behaviour, and their comparison with signatures derived from remotely sensed AET data highlights the strengths and weaknesses of RS information. In a broader context, the remotely sensed products used in this study show significant scatter around the flux tower values, signalling caution regarding their

capacity to mimic observed AET behaviours accurately. Therefore, future studies are encouraged to leverage these AET signatures to evaluate remotely sensed products before their adopted and to better extract and improve information for activities such as hydrological modelling.

## Data availability

The actual evapotranspiration data extracted from Ozflux sites and remotely sensed products are available at

https://doi.org/10.5281/zenodo.14226802.

## Author contribution

Hansini Gardiya Weligamage: Conceptualization, Data curation, Formal analysis, Investigation, Methodology, Validation, Visualization, Writing – original draft preparation, Writing – review and editing.

Keirnan Fowler: Conceptualization, Methodology, Funding acquisition, Supervision, Writing – review and editing

Margarita Saft: Conceptualization, Methodology, Funding acquisition, Supervision, Writing – review and editing

Tim Peterson: Conceptualization, Methodology, Funding acquisition, Supervision, Project administration, Writing – review and editing

Dongryeol Ryu: Conceptualization, Methodology, Supervision, Writing – review and editing

Murray Peel: Conceptualization, Methodology, Funding acquisition, Supervision, Project administration, Writing – review

and edit

## Competing interests

The authors declare that they have no conflict of interest.



**Acknowledgements**

This research was funded by the Australian Research Council (ARC) Linkage Project (LP180100796) with partner organisations Victorian Department of Environment, Land, Water and Planning (DELWP), and Melbourne Water.

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
