# Peer review of "Characterising evapotranspiration signatures for improved behavioural insights"

_Hydrology and Earth System Sciences, 2024_

## Author Comment (AC1)

Responses to the anonymous referee #1 on HESS-2024-373 '*Characterising evapotranspiration signatures for improved behavioural insights*'

Citation: https://doi.org/10.5194/hess-2024-373-RC1

| # | Remarks to author | Authors' responses |
|---|---|---|
| **General comment** | | |
| 1 | The study uses different statistical metrics (referred as hydrologic signatures) like annual median, coefficient of variations (at different timescales) and use them to compare evapotranspiration derived using two remotely sensed products (MODIS and CMRSET) with observations from 17 flux tower sites across Australia.

While this study reports important biases in remote sensing products with observations, it severely lacks in the interpretation of the comparisons and the application of different metrics. As a result, I would recommend a major revision for the manuscript to be publishable in HESS. | Thank you for your constructive comments on our manuscript.

We find your suggestions are valuable for strengthening the manuscript.

We are happy to improve the manuscript in response to the comments you made below.

We hope you would agree that the revisions will improve the interpretation of aspects of the study that you identified as a problem.

Below we provide detailed responses to each of your comments.

Please note that the red texts within the quotation marks in authors' responses below indicates suggested revisions to the text in the revised manuscript. |
| **Comments** | | |
| 2 | C1: Line 44 – 45: I would disagree with this statement. Understanding changes in AET is a well-researched (and ongoing) subject.
I don't think the right motivation for this paper is that statistical metrics like (annual median, coefficient of variations) have not been used to study AET before. Rather I suggest authors motivation should be on comparing and interpreting the remotely sensed evaporation estimates with Flux tower data, the reasons which can lead to discrepancy between them and the use of hydrologic signatures in understanding those biases. | 1) We understand on your disagreement on the statement in L 44-45 as it could imply complete negligence of AET dynamics in research.
Therefore, we suggest revising the sentence replacing the "This lack of attention of AET signatures is surprising …" with "This is a worthy research focus given the importance of AET in the overall water cycle, comprising around 60% of the global terrestrial hydrological cycle"

2) Regarding your statement that characterising AET signatures should not be the motivation of our manuscript, we politely disagree.

We are not claiming that statistical metrics have not been used to study AET. We argue that |

| | | within the hydrological signature space, AET signatures have not yet been systematically defined and collated to quantify different aspects of AET dynamics as mentioned in L40-44 in the original manuscript.

Therefore, the motivation of this study is to characterise a list of statistical metrics (i.e., AET signatures) to quantify AET dynamics as mentioned in L68-70.

We then applied those characterised AET signatures to one of their potential uses, which is assessing the quality of remotely sensed AET products as mentioned in L70-72.

3) We agree that our application of AET signatures to assess the quality of remotely sensed AET is not reflected in the manuscript title.

Therefore, we suggest the following title "Assessing deficiencies in remotely sensed actual evapotranspiration (AET): introducing AET signatures" |
|---|---|---|
| 3 | C2: Line 170-174: It will be useful to also have Morton's equation to estimate potential evaporation written here. | Yes, we agree.

We will include the equation. |
| 4 | C3: Figure 3b: There should be some discussion/explanation about why coefficient of variation (interannual variability) is high in dry regions irrespective of dataset. Does it relate to the interannual variability in rainfall/net radiation/PET over these sites? | Yes, we agree that this needs to be discussed.

In our opinion, AET in dry regions is typically more variable due to high rainfall variability. In addition, PET is generally high, particularly in hot arid environments. Generally, evaporation from soil contributes more to AET than transpiration from vegetation in hot arid environments. Moreover, vegetation tends to be opportunistic during rainfall events and remains dormant during dry periods. Therefore, these factors collectively influence the variability of AET in hot dry regions.

We will discuss this with reference to literature as appropriate. |
| 5 | C4: In addition to coefficient of variation at inter-annual scale, maybe it will be helpful to also compare absolute deviations at annual scale, report it and keep it in supplement. | Sorry, we politely decline your comment for the reasons outlined below. However, we remain open to further consideration if you can elaborate on your reasons to make this comment.

As mentioned above in #2, our motivation in this manuscript is to introduce a new list of signatures (i.e., AET signatures) to the hydrological signature space to characterise AET dynamics. So, this manuscript shows how useful |

| | | these AET signatures can be by comparing remotely sensed AET with flux towers. Therefore, we would prefer to focus on AET signatures.

The comparison of absolute deviations does not add extra information relevant to the utility of AET signatures.

Moreover, the absolute deviation of AET at the annual scale does not allow capturing variability across sites. Instead, it is limited to site specific comparison of variability between remote sensing and flux tower AET. We have outlined this reason in the original manuscript (Section 2.1 L83-85).

Therefore, we do not think that reporting absolute deviations would provide additional insights into the interannual variability.

However, we would be happy to consider this if you could explain what additional information this would add to the AET signature list. |
|---|---|---|
| 6 | C5: Line 213: I am confused about what is meant by CMRSET shows minimal bias? Do you mean spatial variability in lag-12 auto-correlations are low? | Yes, we agree that the sentence may be unclear.

By 'minimal bias', we intended to convey that CMRSET tended to both overestimate and underestimate flux tower periodicity, without showing a clear pattern of over- or under-estimation.
To make the sentence clearer, we suggest a revise text as below.

"Across all sites, the $P_{12month}$ of CMRSET monthly AET does not systematically over- or underestimate flux tower $P_{12month}$. However, there is a considerable scatter meaning some sites showing significant overestimation and others significant underestimation of CMRSET $P_{12month}$ to flux tower $P_{12month}$." |
| 7 | C6: Figure 4: This is an important figure which depicts the difference in seasonality and phase lags in season peaks between remotely sensed data and flux tower observations. But there is no interpretation about what does this imply? My intuition is that this may likely relate to the vegetation parameterizations and surface water stress in remote-sensing derived AET products. However, it is clear that aridity index (defined at long timescales) does not explain these variations either with flux-towers or the remotely sensed data. I suggest authors to look at periodicity and phase lags in surface water-stress if they explain these effects. | Yes, we agree that seasonal scale AET signatures need to be discussed further.

As you mentioned, this may be related to the parameterization of vegetation and surface water stress in remote sensing AET products.

We will discuss this with reference to literature as appropriate.
Regarding the latter part of this comment, we agree that looking into phase lags and water stress are important. That is why we included the signatures on water stress and asynchronicity between AET and PET as described in L93-110 |

| | | and Figure 5b & 5c. Also, we do not see any comment from you regarding those two signatures. Therefore, we assume this addresses your comment. But we would be happy to discuss this further during the later stage of review. |
|---|---|---|
| 8 | C7: Similar to C3, there shall be some discussion/explanation of why coefficient of variation (at monthly scale) shows a variation with aridity. | Yes, we agree that this needs to be discussed.

Similar to our response to your comment C3, we will discuss this with reference to literature as appropriate. |
| 9 | C8: I don't think signature 8 (Index of AET responsiveness to a rainfall event) is a robust metrics. The results presented in figure 6 don't support it either. The response of AET to rainfall will be affected by many confounding factors like water availability, energy availability, land-cover type and seasonality. For e.g, a summer time or winter time rainfall can have very different effects on AET due to differences in net radiation (energy availability). The cloud radiative effects associated with rainfall will also be different across seasons. The presence/absence of vegetation can also significantly alter surface water-stress conditions through water-channelling mechanisms like root systems. A better way to diagnose this effect could perhaps be to first link changes in rainfall to antecedent hydrologic condition like surface water-stress and then look at their response to AET. | Yes, we agree that responsiveness to rainfall can vary depending on antecedent conditions. However, this does not automatically disqualify it as something worth reporting.

This paper focuses on signatures that report discrepancies in AET dynamics, regardless of their underlying causes.

While we agree that discussing the potential causes of these discrepancies is valuable, the primary aim of this manuscript is not just to explain differences between remote sensing and flux tower AET. Instead, our motivation is to define and present AET signatures.

Even if discrepancies arise from different ways, that does not disqualify the AET signature of being reportable. |
| 10 | C9: For each figure, there should be some quantitative measure of consistency like Rsquared or RMSE with respect to observations for both MODIS and CMRSET. This would help assess which dataset performs better for each hydrologic signature. | Yes, we agree.

Thank you for your suggestion. We will update the AET signature figures with a quantitative measure for both MODIS and CMRSET signatures. |
| 11 | C10: It may be useful to analyse if the biases between flux tower observations and remote sensing derived estimates shows a variation with vegetation type for different hydrologic signatures. | Yes, we agree.
We will include a discussion on this, for example by contrasting results across forests, savanna, and grassland ecosystems. |
| 12 | C11: Section 4.2: This section is more of a repetition/summary of results rather than insights. | Yes, we agree.

In line with the other reviewer suggestion, we will improve this section with this comment, which was also supported by the other reviewer who highlighted the same point with some suggestions for the improvements.

We plan to improve this section by expanding the AET signature results related to flux tower AET, which already discussed under this section, comparing them with other studies that have examined AET in Australia. Additionally, we will include a discussion on the reliability of flux |

| | | tower data and their quality assurance techniques. |
|---|---|---|
| 13 | C12: Line 334 – 335: This is not demonstrated in the manuscript rather argued qualitatively. Refer to comment C9. | Yes, we agree.

As responded to the C9, we will include a quantitative measure for CMRSET and MODIS signatures, as it will help assess the dataset performances quantitatively. |
| 14 | C13: Line 351: The current version of the study compares hydrological signatures but does not provide comprehensive insights into AET dynamics. | Thank you for this comment.

We hope that, with the additional discussion as suggested in your comments C3, C7, C10, C11, as well as reviewer 2's suggestions, the revised manuscript will offer more comprehensive insights into AET dynamics. |
| **Minor** | | |
| 15 | Line 19: $AET_{Rs}$ instead of $RS_{AET}$ to be consistent. | Yes, we agree.

Thanks for noticing. We will correct. |
| 16 | Line 213: suggest to change "minimal" to "reduced" | We will change the sentence in response to the C5 above. |
| 17 | Figure 7: Legend missing for MODIS AET | Yes, we agree.

Thank you for noticing.
We will include the missing legend. |
| 18 | For all the figures it may be helpful to have a legend depicting color scale of aridity index (humid – blue, arid – red) or an arrow beside the colormap. | Yes, we agree.

Thank you for this suggestion.
We will indicate the aridity index on the signature figures. |